**communications** engineering

# OpenDosimeter: Open hardware personal X-ray dosimeter

Norah Ger[1], Alice Ku[2], Jasmyn Lopez[2], N. Robert Bennett[2], Jia Wang[3], Grace Ateka[4], Enoch Anyenda[5], Matthias Rosezky[6], Pamela Kilavi[7], Adam S. Wang[2] & Kian Shaker [2,8] ✉

Radiation workers need accurate monitoring of X-ray exposure, but existing solutions are either inaccessible, expensive, or provide delayed feedback. We present OpenDosimeter ([www.opendosimeter.org](www.opendosimeter.org)), an open hardware solution for real-time personal X-ray dose monitoring. Using a scintillator-based X-ray sensor on a custom board powered by a Raspberry Pi Pico, OpenDosimeter provides real-time feedback (1 Hz), data logging (10 hours), and battery-powered operation. A core innovation is that we calibrate the device using $^{241}$Am found in ionization smoke detectors. Specifically, we use the $\gamma$-emissions to spectrally calibrate the dosimeter, then calculate the effective dose from X-ray exposure using the scintillator absorption efficiency and energy-to-dose coefficients derived from public tabulated data. We demonstrate that this transparent approach enables dose rate readings with linear response between 0.1–1000 $\mu$Sv/h at ± 25% accuracy, tested for energies up to 120 keV. The maximum dose rate readings are limited by pile-up effects when approaching count rate saturation (~77,000 counts per second at ~13 $\mu$s average pulse-processing time). The total cost for making an OpenDosimeter is <\$100, which, combined with its open design, enables cost-effective local reproducibility on a global scale. This paper complements the open-source documentation by explaining the underlying technology, algorithms, and areas for future improvement.

Monitoring occupational X-ray exposure is critical for ensuring the short- and long-term safety of radiation workers. To minimize health risks, regulations set dose limits for these workers, such as a yearly effective dose below 20 mSv (averaged over five consecutive years) and not exceeding 50 mSv in any single year[1]. However, while there is limited data on the global availability, many workers in low-resource settings have little or no access to personal dosimeters[2]. Traditional dosimeters can be prohibitively expensive and require complicated logistics around calibration, readout, and reporting. All of this leads to radiation workers having restricted insight into their own exposure levels.

Personal dosimeters are categorized as either active or passive devices. On the passive side, dosimeters based on thermoluminescence (TLDs) or optically stimulated luminescence (OSL) allow for cost-effective scalability and are often offered as a subscription service by an external party[3]. However, individuals wearing these dosimeters receive infrequent feedback on their radiation exposure, typically only monthly or quarterly. In addition, the logistics of collecting dosimeters and sending them to remote facilities

for reading are challenging. Moreover, the lack of immediate feedback hinders the individual in maintaining effective radiation safety practices.

On the active side, there are real-time electronic personal dosimeters[4]. An example is the RaySafe i3 marketed towards interventional radiology procedure,s where dose exposure to the operators can be high, which makes direct feedback critical. Unfortunately, these active devices are often prohibitively expensive (>\$1000) for scaling to larger groups of radiation workers.

Numerous open hardware projects for radiation detection are publicly available, including those using Geiger–Müller tubes (e.g., RadiationD-v1.1[5], uRADMonitor[6]) and solid-state sensors (e.g., OpenGeiger[7], Open Gamma Detector[8], LABDOS01[9]). However, to our knowledge, no open hardware project has yet addressed the critical challenge of accurately calculating the effective dose (in Sieverts) from external X-ray exposure.

We present OpenDosimeter ([www.opendosimeter.org](www.opendosimeter.org)), an open hardware solution for low-cost and real-time personal radiation monitoring. This manuscript complements the open-source documentation (see

[1]Mama Lucy Kibaki Hospital, Nairobi County, Kenya. [2]Department of Radiology, Stanford University, California, US. [3]Department of Environmental Health and Safety, Stanford University, California, US. [4]Secondary Standards Dosimetry Laboratory, Kenya Bureau of Standards, Nairobi, Kenya. [5]Department of Diagnostic Imaging and Radiation Medicine, University of Nairobi, Nairobi, Kenya. [6]NuclearPhoenix, Vienna, Austria. [7]School of Computing and Engineering Sciences, Strathmore University, Nairobi, Kenya. [8]Present address: Department of Applied Physics, KTH Royal Institute of Technology, Stockholm, Sweden. ✉e-mail: kiansd@kth.se

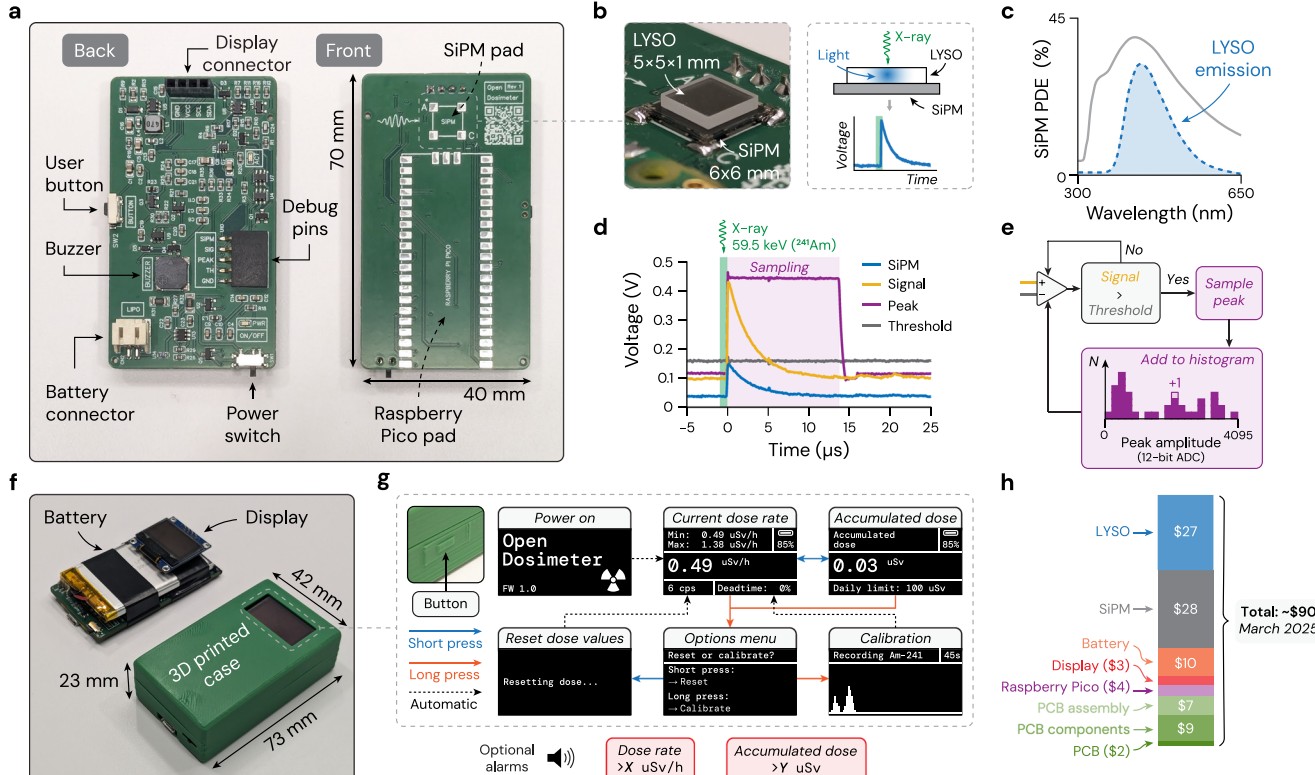

**Fig. 1 | Device overview. a** OpenDosimeter custom board as assembled and delivered by a printed circuit board (PCB) manufacturer. **b** X-ray sensor: lutetium-yttrium oxyorthosilicate (LYSO) crystal mounted on a silicon photomultiplier (SiPM). **c** SiPM photon detection efficiency overlapping with the LYSO emission spectrum (arbitrary y-axis). **d** Signal processing: raw SiPM pulses (blue), amplified signal (yellow), and peak signal amplitude (purple) sampled by the Raspberry Pi Pico (shaded purple); ~13 $\mu$s typical sampling time. **e** Histogram generation from triggered sampling of peak amplitudes whenever the signal exceeds the threshold. **f** Fully assembled OpenDosimeter, without (left) and with (right) the case. **g** User interface: single-button navigation with short/long presses. **h** Component cost breakdown (as of March 2025).

our GitHub repository) by explaining the underlying technology, the algorithms for spectral calibration and dose calculation, as well as benchmarking against a commercial active dosimeter (RaySafe i3).

## Results

### Device overview

The core of OpenDosimeter is the custom-designed printed circuit board (PCB, Fig. 1a). The board can be ordered, assembled, and delivered from any PCB manufacturer (see the GitHub repository), ready for integration with the external components. The back side has analog circuitry for signal processing, connectors for peripherals (display, battery), a buzzer for sound, a button for user interaction, a power switch, and a set of breakout pins to access the analog signals for debugging. The front side has soldering pads for a Raspberry Pi Pico (the microcontroller powering the device) and for a silicon photomultiplier (SiPM). For X-ray detection, we mount a lutetium-yttrium oxyorthosilicate (LYSO) crystal on the SiPM using an optical couplant (Fig. 1b). We chose LYSO as it is non-hygroscopic and relatively easy to source, although other scintillators could be used with different performance trade-offs. X-ray photons absorbed by the LYSO crystal generate visible light centered in the blue region, matching the photon detection efficiency (PDE) of the SiPM (Fig. 1c). We note that the LYSO + SiPM combination is often used for $\gamma$-detection (e.g., in positron emission tomography[10,11]) because of this spectral match, with care taken to subtract the weak background radioactivity of the $^{176}$Lu in the crystal ($\gamma$-emissions at 88, 202, and 307 keV)[12]. We chose a slightly smaller LYSO crystal area (5 × 5 mm) compared to the SiPM (6 × 6 mm) to ensure uniform light detection by the SiPM regardless of the absorption event location in the LYSO crystal. Since we designed OpenDosimeter mainly for occupational exposure to X-rays in diagnostic and interventional radiology settings, where most X-ray

exposure is below 140 keV, we chose a 1 mm LYSO thickness as it has sufficient absorption efficiency in this energy range (see Fig. 2c). Optionally, the LYSO crystal can be coated with a reflective material (e.g., aluminum tape) to guide more light into the SiPM per absorption event. This changes the dynamic range of detectable X-ray energies (cf. pulse height histogram in Fig. 2b) by allowing lower X-ray energies to be detected above the noise floor at the expense of a lower maximum energy threshold defined by pulse height saturation.

Once X-ray photons are absorbed in the LYSO crystal, the resulting burst of blue light is detected by the SiPM, generating current then converted to a voltage pulse. SiPMs are sensitive to temperature variations (gain: −0.8%/°C for our model), so we implemented a voltage bias temperature correction circuit to mitigate this effect (now reduced to −0.25%/°C). The OpenDosimeter board includes dedicated circuitry to amplify the SiPM output, which we henceforth refer to as the "signal" (Fig. 1d). This signal is then processed through an analog peak detector circuit, which holds the amplitude of the signal peak for the Raspberry Pi Pico to sample. Sampling is triggered only when the signal exceeds a threshold level determined by a comparator (Fig. 1e). We program this threshold so that it is dynamically adapted (within the 0–500 mV range) to the signal baseline using an RC-filtered pulse-width modulation (PWM) output from the Pico. When the threshold is exceeded, the Pico samples the peak amplitude and adds it to a histogram of peak amplitudes. Since these peak amplitudes are proportional to the X-ray energies, the histogram represents the detected X-ray spectrum, albeit not yet spectrally calibrated. Once calibrated, this spectrum will serve as the basis for dose rate calculations (cf. next section).

The complete OpenDosimeter is housed in a 3D-printed case measuring 73 × 42 × 23 mm (Fig. 1f), with its compact size making it wearable on the chest or belt for monitoring dose from occupational X-ray exposure.

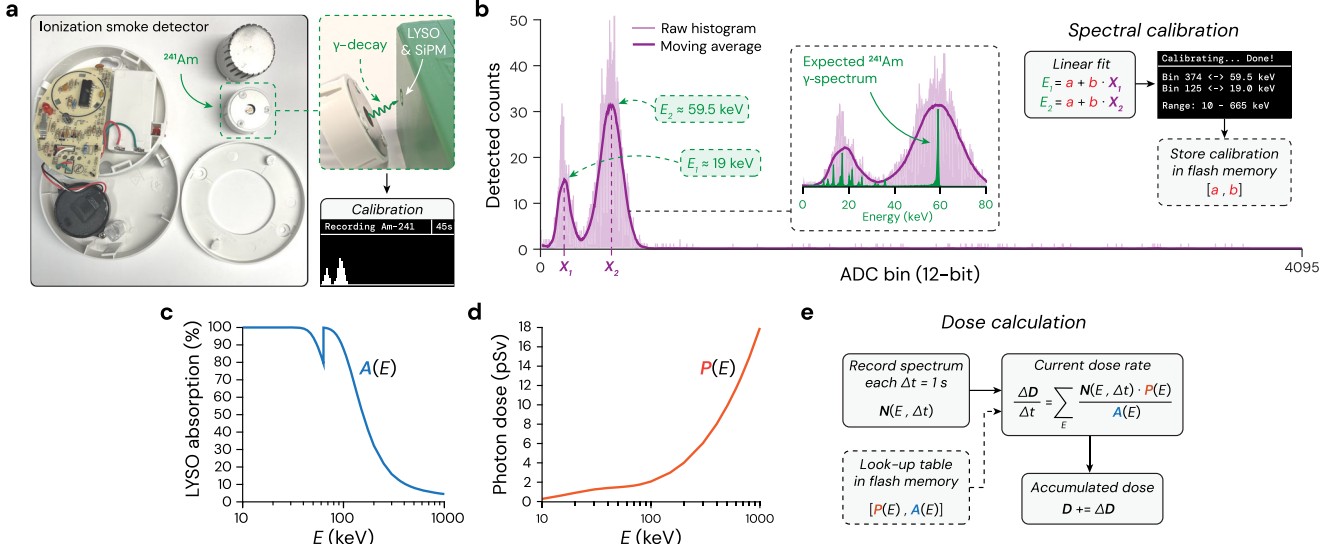

**Fig. 2 | Calibration and dose calculation. a** Disassembled ionization smoke detector with ²⁴¹Am used for spectral calibration **b** Raw calibration histogram (light purple; shaded) and moving average (dark purple; solid) showing two peaks. Inset: zoom on the two peaks with the underlying ²⁴¹Am γ-spectrum overlaid (measured with a CdTe spectrometer). The spectral calibration performs a linear fit of the two peaks with the expected ²⁴¹Am emission energies. **c** Lutetium-yttrium oxyorthosilicate (LYSO, 1 mm thick) spectral X-ray absorption efficiency. **d** Per-photon-dose spectral coefficients, derived from the International Commission on Radiological Protection (ICRP) publication number 116 (Table A.1; anteroposterior direction) and scaled with the LYSO crystal area (25 mm²). **e** Algorithm for calculating the dose rate and accumulated dose.

The device firmware implements a simple state machine for user interaction (Fig. 1g). Upon power-on, the device enters the main operational mode. Short button presses cycle through two display states: showing either the current dose rate or the accumulated dose. The options menu is entered through a long button press, where a short press resets the dose values, and a long press brings the device to the calibration procedure (see next section). Dose values are continuously logged at 1 Hz, with values for the past 10 hours can be accessed and visualized through our Web Interface. The interface does not require any software installation on the user's end, and can be accessed simply by connecting the OpenDosimeter over USB to any computer with an internet connection. The USB connection is also used to charge the lithium-ion polymer (LiPO) battery. A battery with 1200 mAh capacity currently enables up to 20 hours of operation per full charge (power consumption typically <70 mA @ 3.7V).

Lastly, Fig. 1h shows a cost breakdown of the OpenDosimeter. More than half of the total component cost (~$90 as of March 2025) is from the X-ray sensor (LYSO + SiPM). In comparison, commercial dosimeters with similar functionality (real-time, battery-powered, logging capabilities) are typically >$1000 (e.g., RaySafe i3, which we benchmark against in the characterization section).

## Calibration and dose calculation

Unlike the conventional calibration of dosimeters using the 662 keV γ-emission from ¹³⁷Cs, which is typically possible only at centralized facilities, we demonstrate that the OpenDosimeter can be calibrated with any ²⁴¹Am source (e.g., those found in ionization smoke detectors, cf. Fig. 2a). Specifically, we use the low-energy γ-emissions from the source to spectrally calibrate the device. Using tabulated values for the LYSO X-ray absorption efficiency and energy-to-dose conversion coefficients, we then calculate the dose rate without requiring explicit calibration against a known dose rate reference.

Figure 2b shows an example of a calibration histogram of ²⁴¹Am acquired with an OpenDosimeter. The histogram recording stops when the maximum count reaches a predefined limit (e.g., 50 counts) to ensure the acquisition finishes within a reasonable time (e.g., 60 seconds). Next, we perform a moving average on the raw histogram to facilitate the identification of the ADC bins corresponding to the two major peaks observed. To identify the energies of these peaks, we overlay

the expected ²⁴¹Am γ-emissions (Fig. 2b; inset). The peak at the higher ADC bin clearly corresponds to the 59.5 keV emission line, while the peak at the lower ADC bin likely represents the envelope of the low-energy emission lines in the range of 10-23 keV. We empirically assume that the peak of this envelope is around 19 keV, noting that an error in this assumption leads to an error in the spectral calibration and thus adds uncertainty in the dose calculation (cf. next paragraph). Lastly, we assume a linear relationship between the ADC values (i.e., signal peak amplitudes) and X-ray energy, and find the corresponding linear coefficients (Fig. 2b; far right). This calibration procedure also defines the spectral range for which the device can reliably attribute the energy of detected X-ray photons. From a user perspective, this entire process is performed automatically and takes approximately 1 second once the ²⁴¹Am calibration histogram has been recorded, without needing any intervention or parameter tweaking. The calibration parameters [a, b] are then stored in the device flash memory and thus accessible after power cycling.

Once the device has been spectrally calibrated, we use a simple and transparent algorithm for calculating the effective dose (in Sv) from the detected X-ray spectrum. First, we correct the detected spectrum for the spectral absorption efficiency of the 1-mm thick LYSO crystal (Fig. 2c). Next, we multiply the corrected spectrum by per-photon-dose spectral coefficients (Fig. 2d) derived from tabulated data in the International Commission on Radiological Protection (ICRP) publication number 116[13]. We chose the specific coefficients assuming the device will be worn on the chest (ICRP 116; Table 1; anteroposterior direction) and scale these with the LYSO detection area (5 × 5 mm²), resulting in the graph in Fig. 2d. Data from both graphs (Fig. 2c, d) are stored in the Raspberry Pi Pico flash memory and accessible as look-up tables. Lastly, we calculate the current dose-rate at each time point (Δt = 1 s, deadtime corrected) using the detected X-ray spectrum and the look-up tables, and the accumulated dose by incrementally adding the dose per time point (Fig. 2e).

## Characterization and performance demonstration

To characterize the OpenDosimeter performance, we conducted a series of experiments and benchmarked it against a commercial active dosimeter (RaySafe i3). Experiments were performed at X-ray energies up to 120 keV

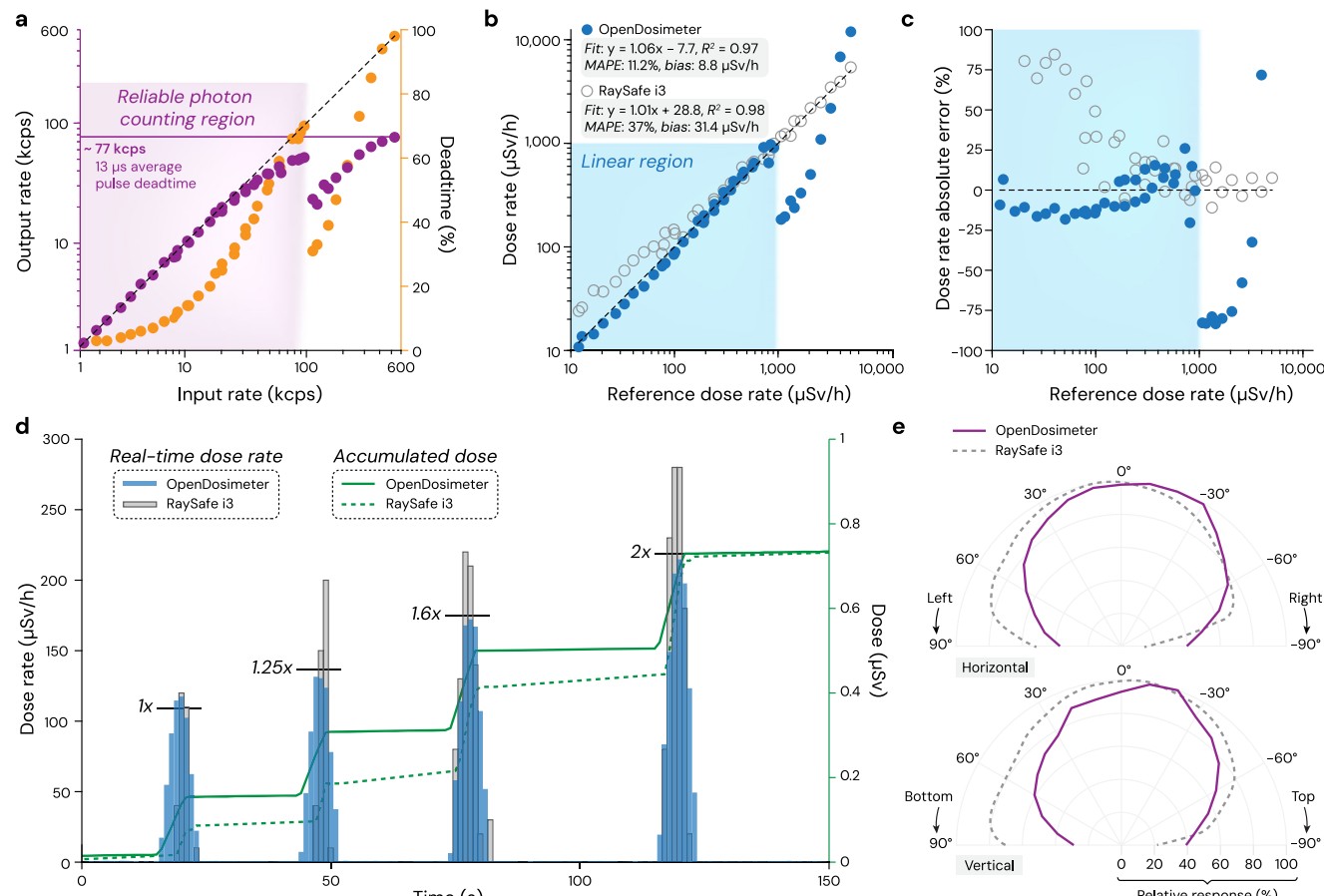

**Fig. 3 | Characterization and performance demonstration with a clinical X-ray tube. a** Output count rate in thousand counts per second (kcps; purple; left) and estimated deadtime (orange; right) as a function of input count rate. **b** Output dose rate of the OpenDosimeter (blue) and the RaySafe i3 reference dosimeter (grey) plotted against the reference dose rate (RaySafe X2 survey meter). Within the linear region, a linear fit to the data, as well as the mean absolute percentage error (MAPE) and the average bias compared to the theoretical 1:1 response (dashed line,) is shown. **c** Relative errors in dose rate measurements. **d** Demonstration of real-time dose rate performance (blue/grey) and accumulated dose (green). **e** Relative response in the angular direction.

using a clinical X-ray tube to simulate the use of OpenDosimeter in a clinical radiology department for personal monitoring of occupational radiation exposure.

First, we evaluated the photon count rate performance of the Open-Dosimeter (Fig. 3a). The output count rate increases linearly with the input rate up to approximately 100 kcps (thousand counts per second), defining a reliable photon-counting region (shaded purple) with a typical non-paralyzable response. Beyond this point, the output becomes unreliable, likely due to the increasing probability of pulse pile-up effects. The maximum output count rate plateaus at ~ 77 kcps, corresponding to a deadtime fraction approaching 100%. The latter is estimated within the Open-Dosimeter by multiplying the output count rate by the average per-pulse sampling time (cf. Fig. 1d, ~13 μs), which ultimately defines the maximum output count rate. Within the reliable photon-counting region, the estimated deadtime percentage can be used as a correction factor for the output count rate.

The dose rate response associated with the data in Fig. 3a of the OpenDosimeter is presented in Fig. 3b, where the output is compared with the RaySafe i3 commercial dosimeter and plotted against the reference dose rate (using a RaySafe X2 survey meter). The reliable photon counting region is clearly mapped to a dose range with linear response extending up to ~ 1 mSv/h (deadtime corrected). Beyond this range, the output becomes unreliable due to the pile-up effects nearing the count rate saturation (cf. Fig. 3a). Compared to the RaySafe i3, the OpenDosimeter performs better at low dose rates, yet the RaySafe i3 has a linear response at the higher dose rates (up to 1 Sv/h according to its datasheet).

To evaluate the accuracy of the OpenDosimeter dose-rate estimates under our test conditions, Fig. 3c shows the relative errors compared to the reference device corresponding to the measurements in Fig. 3b. This shows that our dose calculation algorithm, using only [241]Am for spectral calibration, results in dose rates within ± 25% accuracy of the reference dose rate within its reliable region, well within range of the commercial dosimeter. We plan to conduct a statistical analysis across multiple devices for the next OpenDosimeter version to quantify manufacturing variability and further characterize measurement accuracy.

Next, we demonstrate the real-time performance of the Open-Dosimeter in a dynamic radiation environment compared with the commercial dosimeter (Fig. 3d). The dosimeters were simultaneously exposed to X-rays at increasing tube currents, indicated by multiplicative factors noted as horizontal lines in the graph. Each exposure lasted 5 seconds. The plot shows both the instantaneous dose rate and the accumulated dose as measured by both devices. OpenDosimeter closely tracks the commercial device in both dose rate response and accumulated dose, showcasing the ability to accurately monitor rapidly changing radiation levels.

Lastly, the angular response of the OpenDosimeter is shown in Fig. 3e with the RaySafe i3 (from datasheet) overlaid for comparison. This angular dependence stems from the asymmetric geometry of our LYSO crystal (5 × 5 × 1 mm) with reduced absorption efficiency for X-ray photons entering at oblique angles. For improving dose response to scattered radiation, a more symmetric crystal geometry (e.g., 5 × 5 × 5 mm) should provide better omnidirectional response at the cost of reduced light collection efficiency for absorption events occurring farther from the SiPM.

## Discussion

OpenDosimeter represents a significant advancement in cost-effective personal dosimetry, suitable for distributed manufacturing and local calibration with direct access to dose exposure readings. Our results demonstrate that this open-source device offers performance comparable to commercial dosimeters that are >10× more expensive, in operational conditions relevant for occupational exposure in radiology departments. The current design is suitable for common diagnostic and interventional radiology energy ranges (up to 140 keV), while future versions could be adapted for the MeV energies encountered in radiation therapy, mainly by using a thicker scintillator. Most importantly, the open design allows maximum reproducibility on a global scale.

In particular, we show that by using $^{241}$Am for spectral calibration together with our transparent dose calculation approach, we can achieve surprisingly accurate values of the effective dose (±25% within its dynamic range, cf. Fig. 3c). We note that our accuracy assessment reflects relative error under our test conditions rather than a comprehensive uncertainty budget that would include all systematic effects. Furthermore, its real-time readout and data logging meet the standards set by state-of-the-art commercial active dosimeters at a component cost of <\$100. Despite these strengths, there are several areas where improvements are underway in upcoming upgrades of the OpenDosimeter:

### Power consumption

The current version operates for up to 20 hours on a full charge, drawing <70 mA @ 3.7V with a 1200 mAh battery. While this is sufficient for 2–3 work days, there is significant room for improvement. By optimizing the choice of electronic components on the OpenDosimeter board, it should be easily possible to extend the battery life by at least >2×.

### Integration

To create an even more integrated and compact design, future versions will incorporate the microcontroller chip directly onto the OpenDosimeter board, eliminating the need for soldering a separate Raspberry Pi Pico. Furthermore, the recently released Pico 2 chips (e.g., RP2350) will improve overall performance at a negligible cost increase.

### Dose range

The current reliable dose rate range, up to ~1 mSv/h for energies up to 120 keV, can be sufficient for occupational X-ray exposure in most radiology departments. Some literature reports that typical scatter doses in radiography suites are in the range of 1–2 $\mu$Sv/h[14] (background levels: 0.1–0.5 $\mu$Sv/h), detectable with the OpenDosimeter yet too low for most commercial equivalents (e.g., RaySafe i3 with its detection limit of ~20 $\mu$Sv/h from our studies). We note that the accuracy at these low background levels is difficult to determine due to the limited counting statistics (typically a few counts per second). However, for applications where much higher instantaneous dose rates are expected, such as in interventional radiology, the current performance of OpenDosimeter is insufficient. During interventional procedures with fluoroscopy and radiographic acquisition, instantaneous exposure to the operators can reach 5 mSv/h and 50 mSv/h, respectively[15]. To address this limitation, we are exploring two separate avenues for *hardware* and *software* modifications.

On the *hardware* side, we can improve the dose rate range by using a smaller area LYSO crystal to detect fewer counts at the same dose rates. One can also experiment with thinner LYSO crystals (e.g., 0.5 mm instead of 1 mm) or filtration (e.g., with Al or Cu), and smaller area SiPMs (e.g., 3 × 3 mm or 1 × 1 mm) to further reduce the overall count rate, which will extend the linear range of dose rate measurements. Another benefit of this approach is that it will further reduce the cost. While this approach may reduce sensitivity at low dose rates, our current OpenDosimeter design already measures a count rate of ~2000 cps at a relatively low dose rate of 10 $\mu$Sv/h. This high count rate at low dose rates suggests that we have sufficient

margin to map the dynamic range of the output count rate to a linear dose rate region with a higher maximum value before saturation is reached. With this approach, a 50× improvement of the maximum dose rate (to 50 mSv/h) should be possible.

On the *software* side, a promising approach is to continuously integrate the signal for a so-called "energy-integrating" dose calculation, running parallel to the current photon counting algorithm. This software-based solution aims to enable simultaneous probing of medium to high dose rates (1–50 mSv/h) without hardware changes. Our preliminary tests indicate that the amplified SiPM signal (cf. Fig. 1d) saturates beyond an instantaneous dose rate of 50 mSv/h, suggesting that this is a viable path forward.

To summarize, OpenDosimeter is, to our knowledge, the first open hardware dosimeter capable of accurate, real-time calculations of effective dose. Its innovative calibration procedure, utilizing $^{241}$Am found in ionization smoke detectors, enables local calibration without reliance on specialized facilities. The open design fosters reproducibility and local ownership, thereby supporting global capacity building in radiation safety. As a living project, future software and hardware iterations will continually enhance functionality. Moreover, the open-source licensing (GPLv3) encourages derivative works, which we hope will facilitate local commercialization to address the global demand for personal dosimeters.

## Materials and Methods
### Design, documentation, and reproducibility

The OpenDosimeter GitHub repository contains comprehensive open-source documentation, including detailed instructions for both hardware assembly and software implementation. Additionally, video instructions referenced throughout the documentation can be found on the OpenDosimeter YouTube channel. The data in Fig. 1c for the SiPM photon detection efficiency (PDE) is from the component datasheet (MICROFC-60035, Onsemi), while the LYSO emission profile is from the Luxium Solutions "LYSO Scintillation Material" datasheet. The data in Fig. 1d were acquired by probing the analog signals of an assembled OpenDosimeter (cf. "Debug pins" in Fig. 1a) using a 4-channel oscilloscope (DSOX3024G, Keysight).

### Calibration and dose calculation

To extract samples of $^{241}$Am, we disassembled ionization smoke detectors (cf. Fig. 2a) from various brands (Model 5304, Universal Security Instruments; Model 9120BFF, First Alert; Model i9040, Kidde). While the activity seemed to vary slightly between models, they all performed well for our spectral calibration procedure. The data on the $^{241}$Am $\gamma$-emission in Fig. 1b (inset, green) is from the database accompanying our CdTe spectrometer (X-123CdTe, Amptek). The data in Fig. 2c is from the Luxium Solutions "LYSO Scintillation Material" datasheet. The data in Fig. 2d is from the ICRP 116 report (Table A.1, AP-direction)[13].

### Characterization and performance demonstration

For our reference dose rate measurements, we used the RaySafe X2 system with the survey sensor, which is commonly used for reliable scatter dose survey measurements in radiology departments. X-ray experiments were conducted with a clinical X-ray tube operating with varying voltage (90 or 120 kV), current (20–200 mA, 10 steps), and Pb-filtration (2–8 mm). This resulted in average energies between 50–80 keV (reported by the RaySafe X2 system), depending on tube voltage and Pb-filtration thickness, incident on the devices (OpenDosimeter/RaySafe i3/RaySafe X2) placed 1 meter directly in front of the tube. By varying the Pb-filtration/voltage and sweeping over the tube current settings, we were able to gather data points over a range of reference dose rates from approximately 10 $\mu$Sv/h to 5 mSv/h (Fig. 3b, c).

## Data availability

The data presented in this study can be requested from the corresponding author.

## Code availability

The software for the OpenDosimeter is available at the OpenDosimeter GitHub repository. The data shown in this study has been extracted from an OpenDosimeter device using the datalog web interface hosted on the project website (www.opendosimeter.org).

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

## Acknowledgements

The OpenDosimeter project is a derivative of the Open Gamma Detector project[8]. We thank Shirin Pourashraf at Stanford University for assisting with SiPM configurations. We acknowledge funding support from the Knut and Alice Wallenberg Foundation and the King Center for Global Development at Stanford University.

## Author contributions

N.G. and K.S. conceived the project. N.G., A.K., A.S.W., and K.S. designed the device and its functionality. G.A., E.A., and P.K. gave input on functionality and user design. M.R. and K.S. developed the hardware and software. N.R.B., J.W., and K.S. performed the characterization. K.S. analyzed the data. J.L. and K.S. wrote the open-source documentation. K.S. led the project and wrote the manuscript with input from all authors.

## Funding

## Competing interests

The authors declare no competing interests.

## Additional information

**Peer review information** : *Communications Engineering* thanks Yunlong Wang and the other anonymous reviewers for their contribution to the peer review of this work. Primary Handling Editors: [Rosamund Daw]. Peer review reports are available.

