## [Transparent Peer Review file · Communications Engineering]

OpenDosimeter: Open Hardware Personal X-ray Dosimeter

Corresponding Author: Dr Kian Shaker

Version 0:

Reviewer comments:

Reviewer #1

(Remarks to the Author)

This work introduces a low-cost, real-time personal x-ray dosimeter. After reviewing this study, I found it to be both novel and comprehensive. I have a few minor suggestions for further improvement:

1. The authors should provide a detailed explanation of the physics behind the dosimeter, specifically how it detects radiation doses.
2. For healthcare applications, the dosimeter should be tested in a cancer center with various treatment units or medical imaging modalities, such as medical linear accelerators, computed tomography, gamma knife, cyber knife, etc. Using a source from a smoke detector does not seem sufficiently rigorous.
3. The dosimeter was only tested with kV photon beams. However, MV photon beams are also widely used in radiation medicine and should be considered.
4. The display in Figure 3e is incomplete.
5. How does the dosimeter respond to other particle beams, such as electrons and protons?

Reviewer #2

(Remarks to the Author)

The authors developed an open-source hardware-based X-ray personal dosimeter, which is very interesting and meaningful work. I recommend this manuscript, published in communications engineering, however, some issues should be clarified or discussed.

1. In fact, some commercial personal dosimetric devices, especially pen-shaped Geiger-Müller counters, have been available even on online sellers. Can the authors emphasize the improvement of their devices besides the open-source hardware?
2. Can the devices record the background doses? How accurate is it?
3. Would placing mirrors around the crystal increase sensitivity?
4. Are polymer scintillators suitable for this device?
5. What are the main factors to consider when choosing between LSO and LYSO crystals?
6. The number of references does not seem consistent with the journal's conventions.

Reviewer #3

(Remarks to the Author)

The authors present a scintillator/SiPM-based personal x-ray dosimeter for low energy x-ray beams. Although the concept of such a detector is not new, this study successfully implemented this detector for dose monitoring applications. There are several issues that need to be addressed. The organization of the manuscript can be greatly improved. The methods section needs to be before the results section for better readability. A big portion of the results section describes the methodology used in this study and can be merged with the methods section. The authors need to compare the results in this study with existing literature on real-time dosimeters for radiation protection (<https://doi.org/10.1016/j.radmeas.2021.106607>, <https://doi.org/10.1016/j.radmeas.2011.07.006>, <https://doi.org/10.1007/s40042-021-00075-5>, <https://doi.org/10.1016/j.ejmp.2016.11.006>). The authors mention the RaySafe dosimeter but there are other dosimeters, such as CMOS-based, used for the same application. The authors emphasize the cost of the dosimeter, which is relevant and a benefit of the proposed detector, but it is not appropriate to compare the construction cost of one detector to the selling price of another. A comparison of the construction cost of RaySafe system will be more relevant as a discussion point. It is difficult to discern the scientific value added by the pilot study. Although conducting a pilot study is a good idea, it takes away from the main scope of the manuscript.

I have some minor comments as well:

- Page 2: "where X-ray exposure rarely exceed energies of 140 keV". This is not true for radiotherapy where high energy x-rays are very common.
- Figure 3b: Needs an R² value
- Page 5: "Compared to the RaySafe i3, the OpenDosimeter performs better at low dose rates, yet the RaySafe i3 has a linear response at the higher dose rates". This statement should include some quantitative metrics such as R² or other fit metrics.
- Page 5: "To quantify the uncertainty of the OpenDosimeter dose-rate estimates". The uncertainty reported in this paper is Type A (please refer to NIST uncertainty guidelines). As shown by the author, the proposed dosimeter exhibits temperature and angular dependence, which needs to be accounted in the uncertainty budget. I surmise that the actual uncertainty is much larger than this.
- Figure 3e: there is significant angular dependence exhibited by the dosimeter. The authors need to expand on this in the results section and the ramifications of this dependence. Since x-ray personal dosimetry mostly involves scattered photon radiation, the response of the detector will vary greatly with the angle of incidence.

Version 1:

Reviewer comments:

Reviewer #1

(Remarks to the Author)

The authors have addressed all my concerns.

Reviewer #2

(Remarks to the Author)

The revision made by the authors is satisfactory.

Reviewer #3

(Remarks to the Author)

The authors did not improve the manuscript and brushed away most of the concerns made in the previous review cycle.

Major comments:

- The comment about comparing the proposed dosimeter with existing systems will add a valuable perspective for the audience and acknowledge the existence of other systems. I am not recommending that the authors compare all of their findings with these other works - just simply discussing the advantages and disadvantages of their system compared to the existing ones.
- I am still unsure on the relevance of the pilot study. Majority of the data provided here is subjective.
- The authors refuse to perform quantitative statistical analysis for their data (especially figure 3a and 3b)
- The authors failed to expand on the angular dependence of their dosimeter, which appears to have large source of error.

Dear Editor and Reviewers,

We thank you for providing constructive feedback and thoughtful comments, which we used to improve our manuscript. Below is our point-by-point response (**blue**) including modifications to the revised manuscript (**red**) according to the reviewer comments (**black**). Line numbers referred to in this response relate to the new revised manuscript.

In addition to the responses to the Reviewer comments below, we added the following in the revised manuscript:

- Section on “Data availability”
- Section on “Code availability”

Yours sincerely,
The authors

Reviewer #1

This work introduces a low-cost, real-time personal x-ray dosimeter. After reviewing this study, I found it to be both novel and comprehensive. I have a few minor suggestions for further improvement:

1. The authors should provide a detailed explanation of the physics behind the dosimeter, specifically how it detects radiation doses.

Response: We believe that we have sufficiently described how the dosimeter works in terms of detecting X-ray photons (Fig. 1, and section on “Device overview” in “Results”) and how it is calibrated for converting the detected X-ray photons to their dose contribution (Fig. 2, and section on “Calibration and dose calculation” in “Results”). Thus we feel that no additions are needed here.

2. For healthcare applications, the dosimeter should be tested in a cancer center with various treatment units or medical imaging modalities, such as medical linear accelerators, computed tomography, gamma knife, cyber knife, etc. Using a source from a smoke detector does not seem sufficiently rigorous.

Response: Thank you for the suggestion. However, we believe this is out of the scope of our study, which is focusing on the design and construction of an open hardware personal dosimeter for the X-ray range of up to ~140 keV (meaning, not suitable for MeV beams in radiation therapy). The smoke detector is used for local and accessible spectral calibration (Fig. 2), although we tested the performance of the device using a clinical X-ray tube for energies up to 120 keV (Fig. 3). This is stated in the first paragraph of the “Characterization and performance demonstration” section in “Results”.

To clarify the latter, we slightly modified the caption of Fig. 3:

“FIG. 3: Characterization and performance demonstration with a clinical X-ray tube”

3. The dosimeter was only tested with kV photon beams. However, MV photon beams are also widely used in radiation medicine and should be considered.

Response: We do not make any claims that the OpenDosimeter is suitable for MV photon beams, and make it clear (e.g., in the abstract) that the performance has been tested up to 120 keV. The absorption efficiency of our 1-mm thick LYSO crystal is shown in Fig. 2 c, which clearly indicates that one would need a different design (mainly a thicker scintillator) for adapting the OpenDosimeter for MeV energies.

To clarify that the OpenDosimeter design could be extended to MeV energies, we added the following sentence in the “Discussion” section:

“The current design is suitable for common diagnostic and interventional radiology energy ranges (up to ~140 keV), while future versions could be adapted for the MeV energies encountered in radiation therapy, mainly by using a thicker scintillator.”

4. The display in Figure 3e is incomplete.

Response: We do not understand this comment. Fig. 3e is complete as we intended (see screenshot below), showcasing the angular response of the device to incident X-ray radiation. Note that the angles shown here are incidence angles of the radiation to the dosimeter, with angles larger than $\pm 90^\circ$ indicating irradiation on the back side of the dosimeter (which is irrelevant as the dosimeter is intended to be worn with its back towards the chest).

5. How does the dosimeter respond to other particle beams, such as electrons and protons?

Response: The OpenDosimeter is specifically designed for X-ray photon detection and has not been characterized for other particle types like electrons or protons. The LYSO

scintillator and SiPM combination is optimized for the energy range of diagnostic and interventional X-rays (up to ~140 keV). Response to other particles would require different calibration procedures and potentially different detector materials, which is beyond the scope of this work (X-ray dosimetry).

Reviewer #2

The authors developed an open-source hardware-based X-ray personal dosimeter, which is very interesting and meaningful work. I recommend this manuscript, published in communications engineering, however, some issues should be clarified or discussed.

1. In fact, some commercial personal dosimetric devices, especially pen-shaped Geiger-Müller counters, have been available even on online sellers. Can the authors emphasize the improvement of their devices besides the open-source hardware?

Response: While GM tube-based devices exist and can be broadly found commercially, their ability to calculate effective dose from X-ray spectra is limited because they cannot discriminate between different X-ray energies (i.e., each incident photon count is recorded as a “click”, regardless of energy). Therefore, to our knowledge, GM tube-based devices are not common for personal dosimetry in radiology settings. Other than the open-source hardware, our key improvements include: (1) spectral energy information for accurate dose calculation, and (2) local calibration using widely available ^{241}Am sources.

We believe these points are already clearly covered in our manuscript, warranting no additional text.

2. Can the devices record the background doses? How accurate is it?

Response: Yes, the device can detect dose rates down to background levels. We have observed that background levels (0.1-0.5 $\mu\text{Sv/h}$) are within our detection range and contribute to the accumulated dose over time. However, accuracy at these low rates is primarily limited by counting statistics, and even commercial survey meters show reduced accuracy in this range. While we demonstrate $\pm 25\%$ accuracy for dose rates above ~ 10 $\mu\text{Sv/h}$, we refrain from making specific accuracy claims for background-level measurements due to these statistical limitations.

To clarify the latter, we have added a sentence in the “Discussion”:

“We note that the accuracy at these low background levels is difficult to determine due to the limited counting statistics (typically a few counts per second).”

3. Would placing mirrors around the crystal increase sensitivity?

Response: Adding mirrors (e.g., reflective coating) around the LYSO crystal improves the light collection efficiency into the SiPM for a single X-ray detection event (cf. Fig. 1b). This translates into a higher pulse height as detected by the SiPM for the same incident X-ray

energies, and thereby changes the dynamic range in terms of X-ray energies that can be detected. However, this does not change the sensitivity in terms of the detectable dose rate range as the main limitation for this sensitivity is the X-ray photon count rate (cf. Fig. 3a, which stays the same regardless of reflective coating on the crystal).

To discuss the possibility of adding reflective coating on the LYSO crystal, we added the following text in the “Device overview” section under “Results”:

“Optionally, the LYSO crystal can be coated with a reflective material (e.g., aluminum tape) to guide more light into the SiPM per absorption event. This changes the dynamic range of detectable X-ray energies (cf. pulse height histogram in Fig. 2b) by allowing lower X-ray energies to be detected above the noise floor at the expense of a lower maximum energy threshold defined by pulse height saturation.”

4. Are polymer scintillators suitable for this device?

Response: Polymer scintillators require much thicker scintillators for the same stopping power as, e.g., 1 mm LYSO. Although they could likely be adapted to work with the OpenDosimeter (with a change in sensitivity, detection efficiency) we recommend working with LYSO.

5. What are the main factors to consider when choosing between LSO and LYSO crystals?

Response: For our application, both LSO and LYSO work well. LYSO has the advantage of being more widely available and typically cheaper. The key factors are light output, decay time, and density. These factors are comparable between LSO and LYSO for X-ray detection in our energy range.

To add some text on the choice of the scintillator, we added the following in the “Device overview” section of “Results”:

“We chose LYSO as it is non-hygroscopic and relatively easy to source, although other scintillators could be used with different performance trade-offs.”

6. The number of references does not seem consistent with the journal's conventions.

Response: The list of references have increased in the revised manuscript (see below), yet we don't find a recommended number for Communications Engineering, and we have cited work as we find suitable throughout the manuscript.

Reviewer #3

The authors present a scintillator/SiPM-based personal x-ray dosimeter for low energy x-ray beams. Although the concept of such a detector is not new, this study successfully implemented this detector for dose monitoring applications. There are several issues that need to be addressed. The organization of the manuscript can be greatly improved. The

methods section needs to be before the results section for better readability. A big portion of the results section describes the methodology used in this study and can be merged with the methods section.

Response: We respectfully disagree with the Reviewer recommendation to reorganize the manuscript. For Communications Engineering the “Results” section typically appears before “Methods”, and the main results of our work is indeed the design decisions in making the OpenDosimeter. The “Methods” section is instead kept succinct and complements the main resource for reproducibility which is our open source documentation (<https://github.com/OpenDosimeter/OpenDosimeter>).

The authors need to compare the results in this study with existing literature on real-time dosimeters for radiation protection (<https://doi.org/10.1016/j.radmeas.2021.106607>, <https://doi.org/10.1016/j.radmeas.2011.07.006>, <https://doi.org/10.1007/s40042-021-00075-5>, <https://doi.org/10.1016/j.ejmp.2016.11.006>).

Response: For our work we deliberately focused on open hardware projects for radiation detection (cf second to last paragraph in the “Introduction” section). None of the devices in the literature recommended by the Reviewer is open-source, so we find no suitable way of citing these works in our manuscript. Benchmarking the OpenDosimeter against other devices published in literature is beyond the scope of our work, but might be relevant for future studies examining the performance in a realistic clinical setting. Furthermore, we find that the papers suggested by the Reviewer represent a somewhat random selection from the vast broad literature on closed-source radiation detectors.

To emphasize that there is vast literature on both passive and active personal dosimeters, we added the following review papers in the “Introduction” section in the respective paragraphs:

- Yang, Zetian, et al. "Passive dosimeters for radiation dosimetry: materials, mechanisms, and applications." *Advanced Functional Materials* 34.41 (2024): 2406186.
- Ginjaume, M., et al. "Overview of active personal doseimeters for individual monitoring in the European Union." *Radiation protection dosimetry* 125.1-4 (2007): 261-266.

The authors mention the RaySafe dosimeter but there are other dosimeters, such as CMOS-based, used for the same application. The authors emphasize the cost of the dosimeter, which is relevant and a benefit of the proposed detector, but it is not appropriate to compare the construction cost of one detector to the selling price of another. A comparison of the construction cost of RaySafe system will be more relevant as a discussion point.

Response: We respectfully disagree with the Reviewer on the cost comparison. Not only are construction costs of commercial dosimeters typically proprietary information, but importantly, from a user perspective, the construction cost of commercial dosimeters is irrelevant. The construction cost of OpenDosimeter is a fair comparison to the selling price of commercial dosimeters as these are the actual price tags relevant to a user deciding

between the options. Clearly, one of the major advantages of open hardware devices is the cost benefit, which is one of the main points of our manuscript.

It is difficult to discern the scientific value added by the pilot study. Although conducting a pilot study is a good idea, it takes away from the main scope of the manuscript.

Response: The value of the pilot workshop is evidence that the device is indeed reproducible, and is a key metric of open hardware projects.

To clarify the point on reproducibility, we slightly modified the caption of Figure 4:

“FIG. 4. OpenDosimeter **reproducibility as a tool for capacity-building in radiation technology.”**

I have some minor comments as well:

- Page 2: “where X-ray exposure rarely exceed energies of 140 keV”. This is not true for radiotherapy where high energy x-rays are very common.

Response: We clarified our intended design for OpenDosimeter to be used in diagnostic and interventional radiology settings (e.g., where X-ray tubes are the main source of occupational exposure) with the following modification to the text in the “Device overview” section under “Results”:

“Since we designed OpenDosimeter mainly for occupational exposure to X-rays in **diagnostic and interventional radiology settings, where most X-ray exposure is below 140 keV [...]”**

- Figure 3b: Needs an R^2 value

Response: Thank you for this suggestion. Upon careful consideration, we believe adding an R^2 value would be misleading in this context for several reasons: (1) Fig. 3b demonstrates a typical response of a single OpenDosimeter device, and performing a linear fit to warrant an R^2 calculation would not be meaningful. (2) The dose response has two separate regions: a somewhat linear region ending with a discontinuity around ~ 1 mSv/h due to count rate saturation. (3) A single R^2 value would either artificially combine these distinct behaviors or arbitrarily separate them. Instead, for transparency we provide Fig. 3c showing the relative errors, which gives readers a more direct insight into the typical measurement accuracy in its suitable operational range.

To clarify this, we made the following modification in the “Characterization and performance demonstration” section under “Results”:

“The dose rate response **associated with the data in Fig. 3a of the OpenDosimeter is presented in Fig. 3b [...]**”

We also added this sentence at the end of the following paragraph:

"We plan to conduct a statistical analysis across multiple devices for the next OpenDosimeter version to quantify manufacturing variability and further characterize measurement accuracy."

- Page 5: "Compared to the RaySafe i3, the OpenDosimeter performs better at low dose rates, yet the RaySafe i3 has a linear response at the higher dose rates". This statement should include some quantitative metrics such as R^2 or other fit metrics.

Response: Continuing the discussion raised from the point above, we believe the data in Fig. 2b and 2c are transparent enough to clearly support this statement. We refrain from making any statistical fits due the points raised above.

- Page 5: "To quantify the uncertainty of the OpenDosimeter dose-rate estimates". The uncertainty reported in this paper is Type A (please refer to NIST uncertainty guidelines). As shown by the author, the proposed dosimeter exhibits temperature and angular dependence, which needs to be accounted in the uncertainty budget. I surmise that the actual uncertainty is much larger than this.

Response: Thank you for raising the points about uncertainty characterization. What we present in Fig. 3c is the relative error (percentage difference) between our device and a reference dosimeter, rather than a formal statistical Type A uncertainty analysis. This shows the accuracy of a representative device under our specific test conditions, but does not constitute a complete uncertainty budget as you point out.

To clarify this, we revised the corresponding sentences in the "Characterization and performance demonstration" section under "Results":

"To evaluate the accuracy of the OpenDosimeter dose-rate estimates under our test conditions, Fig. 3c shows the relative errors compared to the reference device [...]"

We further added the following sentence in "Discussion" to clarify this point:

"We note that our accuracy assessment reflects relative error under our test conditions rather than a comprehensive uncertainty budget that would include all systematic effects."

- Figure 3e: there is significant angular dependence exhibited by the dosimeter. The authors need to expand on this in the results section and the ramifications of this dependence. Since x-ray personal dosimetry mostly involves scattered photon radiation, the response of the detector will vary greatly with the angle of incidence.

Response: Thank you for this suggestion, and we agree with your point.

To expand the section on angular dependence, we added the following at the end of the

"This angular dependence stems from the asymmetric geometry of our LYSO crystal (5×5×1 mm), which has reduced absorption efficiency for photons entering at oblique angles. For scattered radiation applications, a cubic crystal geometry (e.g., 5×5×5 mm) would provide

better omnidirectional response, though with potentially reduced light collection efficiency for events occurring farther from the SiPM.”

Dear Editor and Reviewers,

Thank you for the comments on our revised manuscript, which we used to improve it further. Below is our point-by-point response (**blue**) including modifications to the revised manuscript (**red**) according to the Reviewer #3 comments (**black**).

In addition, upon revising the Abstract, we added this introductory sentence to give some relevant context to our work:

“Radiation workers need accurate monitoring of X-ray exposure, but existing solutions are either inaccessible, expensive, or provide delayed feedback.”

Yours sincerely,
The authors

Reviewer #3

The authors did not improve the manuscript and brushed away most of the concerns made in the previous review cycle.

Major comments:

-The comment about comparing the proposed dosimeter with existing systems will add a valuable perspective for the audience and acknowledge the existence of other systems. I am not recommending that the authors compare all of their findings with these other works - just simply discussing the advantages and disadvantages of their system compared to the existing ones.

Response: Thank you for clarifying your initial comment. We believe we have adequately discussed the main advantages of our device compared to existing commercial systems (primarily the open-source design and accessible calibration) and acknowledge the existence of other real-time dosimeter systems reported in the literature. Given that most devices reported in the literature are not open source and provide varying levels of performance detail, we find it challenging to expand on specific advantage/disadvantage comparisons without making assumptions about unpublished implementation details. We believe the current manuscript provides readers with a clear understanding of the strengths and limitations of our device relative to the established commercial standard (RaySafe i3) that serves as an appropriate benchmark for this application.

-I am still unsure on the relevance of the pilot study. Majority of the data provided here is subjective.

Response: After thorough discussion among the authors and considering the responses from the previous review round, we have decided to remove Figure 4 and the sections related to the pilot workshop from this manuscript. This allows us to focus solely on the technical aspects of OpenDosimeter. We plan to submit our workshop experiences (previous

Figure 4 and related sections) as a separate “Perspective” or “Comment” piece which will enable us to clearly separate the technical development of OpenDosimeter from its value as an educational tool in radiation safety (based around previous Figure 4).

We believe this decision does not change any of the technical aspects of this manuscript on the OpenDosimeter, and should hopefully make it more streamlined, with our subjective experiences running the workshop to be shared in a separate publication piece. We note that the Reviewer comments in both revision rounds were solely on the technical aspects and not on the workshop content.

-The authors refuse to perform quantitative statistical analysis for their data (especially figure 3a and 3b)

Response: As the Reviewer originally requested, we have now added some quantitative statistical analysis to the data in Figure 3. For the linear region in Fig 3b, we added a linear fit to the OpenDosimeter and RaySafe i3 data, as well as the mean absolute percentage error (MAPE) and the average bias. See new updated Fig 3b below:

And an updated caption:

“FIG. 3. Characterization and performance demonstration. [...] b, Output dose rate of the OpenDosimeter (blue) and the RaySafe i3 reference dosimeter (grey) plotted against the reference dose rate (RaySafe X2 survey meter). Within the linear region, a linear fit to the data as well as the mean absolute percentage error (MAPE) and the average bias compared to the theoretical 1:1 response (dashed line) is shown. [...]”

For Fig. 3a, we are unsure what kind of quantitative statistical analysis would be instructive to the reader. The output count rate of the device shows a behavior characteristic of a non-paralyzable photon-counting detector (as expected) although a more detailed

characterization is left for future work and was outside the scope of our presented work where we simply demonstrate the typical ranges of count rates the device can detect.

To clarify the non-paralyzable nature of the dosimeter, we added this brief text in the manuscript:

“The output count rate increases linearly with the input rate up to approximately 100 kcps (thousand counts per second), defining a reliable photon-counting region (shaded purple) **with a typical non-paralyzable response.**”

-The authors failed to expand on the angular dependence of their dosimeter, which appears to have large source of error.

Response: We acknowledge that we may not have been sufficiently clear about where this discussion was added in the last revision. In response to your previous comment in the last revision round, we did add the following discussion at the end of the "Characterization and performance demonstration" section:

“This angular dependence stems from the asymmetric geometry of our LYSO crystal (5×5×1 mm), which has reduced absorption efficiency for photons entering at oblique angles. For scattered radiation applications, a cubic crystal geometry (e.g., 5×5×5 mm) would provide better omnidirectional response, though with potentially reduced light collection efficiency for events occurring farther from the SiPM.”

We agree that the angular dependence represents a limitation of the current design, particularly for applications involving scattered radiation where photons arrive from multiple angles. We report this characteristic transparently so that users can account for it in their applications and future design iterations can address this limitation through geometric modifications as suggested above.